# Aerodynamics of Cycling Skinsuits Focused on the Surface Shape of the Arms

**Sungchan Hong** [1,2,*] and **Takeshi Asai** [1,2]

1 Faculty of Health and Sport Sciences, University of Tsukuba, Tsukuba 305-8574, Japan; asai.takeshi.gf@u.tsukuba.ac.jp

2 Advanced Research Initiative for Human High Performance, University of Tsukuba, Tsukuba 305-8574, Japan

* Correspondence: hong.sungchan.fu@u.tsukuba.ac.jp; Tel.: +81-29-853-2650

**Abstract:** In cycling, air resistance corresponds to 90% of the resistance on the bicycle and cyclist and 70% of this is applied to the body of the cyclist. Despite research on postures that could reduce air resistance, few studies have been conducted on full-body cycling suits. As the aerodynamics of the surface shape of clothing fabric are still unclear, the airflow around cyclists and air resistance were examined using a computational fluid dynamics (CFD) method and wind tunnel experiment. Specifically, in this study, we focused on how different surface shapes of cycling suit fabrics affect air resistance. CFD results indicate that air resistance during a race was high at the head, arms and legs of the cyclist. In the wind tunnel experiment, a cylinder model resembling the arms was used to compare the aerodynamic forces of various fabrics and the results showed that air resistance changed according to the fabric surface shape. Moreover, by changing the fabric shape of the arms of the cycling suits, reduction of air resistance by up to 8% is achievable. These results suggest that offering the most appropriate suit type to each cyclist, considering race conditions, can contribute to further improvement in their performance.

**Keywords:** aerodynamics; computer fluid dynamics; cycling; fabric; wind tunnel

## 1. Introduction

In bicycle sports, such as track races, a wide range of approaches are introduced to reduce loads such as air and road resistances generated during cycling. Previous study indicates that air resistance is responsible for 90% of the total resistance applied to the bicycle and body of the cyclist during a road race and 70% of that resistance is applied to the body of the cyclist [1]. Additionally, the aerodynamics around the body of the cyclists are said to act as a very complex vortex generator [2,3]. Furthermore, there are basically two ways of reducing this air resistance (drag) generated around the cyclist. The first is to reduce the cross-sectional area by making the posture of the cyclist smaller [4], and the second is to move the separation line by changing the surface shape of the body of the cyclist [5]. In track sports where an average speed of approximately 60 km/h and a maximum speed of over 70 km/h exist, development of aerodynamically superior clothing is a crucial factor to convert the energy into propulsion efficiently. For this reason, the airflow around cyclists and the slipstream have been analyzed, in detail, in previous studies [6–10], while others have focused on methods to evaluate the air resistance acting on bicycles [11]. Lately, researchers have investigated the airflow to the wheels using computational fluid dynamics (CFD) [12], while others have focused on the effect of the posture of a cyclist on the drag and airflow [13–16]. Additionally, a review paper summarizing varied previous studies on bicycle aerodynamics was recently published [17].

However, the aerodynamics studies that analyze the relationship between clothing and air resistance by focusing on cycling fabric are still rare. The data required to propose an optimized suit, considering the elements such as running conditions and body and posture of the athlete, are still scarce especially in the case of single-body skinsuits used for

track races [11]. Therefore, in order to develop a suit designed to enhance the performance of the cyclist at the fabric level regardless of the wearer, it is necessary to acquire further knowledge about the air resistance acting on each part of the cyclist's body using CFD numerical analyses and wind tunnel experiments.

In this study, the air resistance acting on cyclists was analyzed in order to propose a suit that can further reduce air resistance. To this end, parts of the cyclist's body where the load is concentrated were evaluated by numerical analysis and simulation. Special attention was given to the arms, which experience particularly high air resistance according to previous studies [8,13,14]. Therefore, in this study, a full-scale mannequin that simulates a cyclist and a cylinder model in a wind tunnel experiment were used to determine the relationship between the fabric structure of the arms and the air resistance acting on the suit.

## 2. Methods

### 2.1. Wind-Tunnel Experiment

The wind tunnel equipment used in this experiment was a circulating wind tunnel (procured from San Technologies Co., Ltd., Tochigi, Japan) located in the University of Tsukuba (Figure 1). This wind tunnel has a maximum wind speed of 55 m/s, a blower of 1.5 m × 1.5 m, a wind speed distribution within ±0.5% and a disturbance of less than 0.1%. In this basic experiment, a cylinder model simulating an arm was used to examine the relationship between the air resistance acting on the cyclist's arms and the suit (Figure 1). This cylinder model had a diameter of 100 mm and length of 300 mm to simulate the actual shape of a cyclist's arm and the side face of its resin body was covered with suit fabrics of the size equal to this cylinder size. Then, the aerodynamics were measured with wind speeds between 50 and 60 km/h at intervals of 5 km. The forces acting on the cylinder model were measured with a sting-type six-component force detector (LMC-61256, procured from Nissho Electric Works, Tokyo, Japan). In this study, the four fabric types shown in Figure 2 were produced and compared in order to examine the aerodynamic effects of different surface shapes. Here, A has small uneven grooves, B has long and shallow vertical grooves, C has horizontal grooves and D is a smooth type of fabric. The groove lines were spaced 10 mm apart. As basic research on different types of fabrics, this study entailed an investigation of the aerodynamic properties of these four fabric types in the wind tunnel. Although the raw material of the fabrics used in the experiment cannot be disclosed, they were all made of the same material.

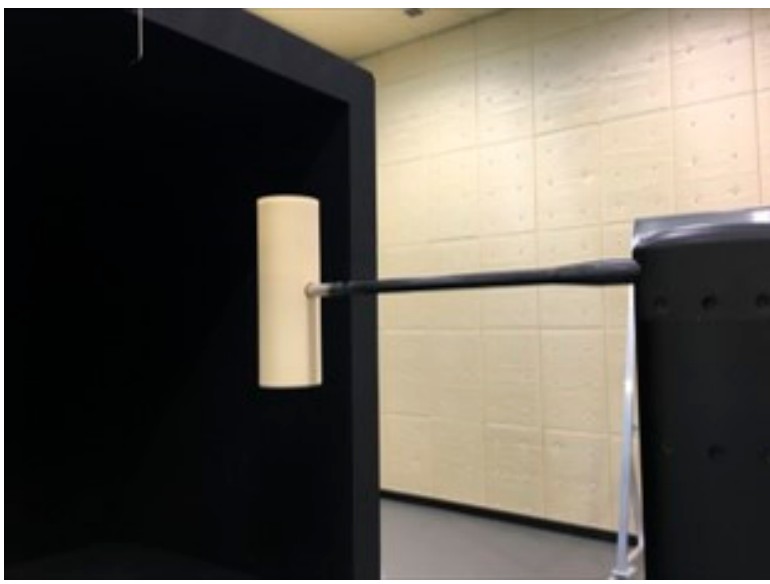

**Figure 1.** Wind-tunnel setup (cylinder model).

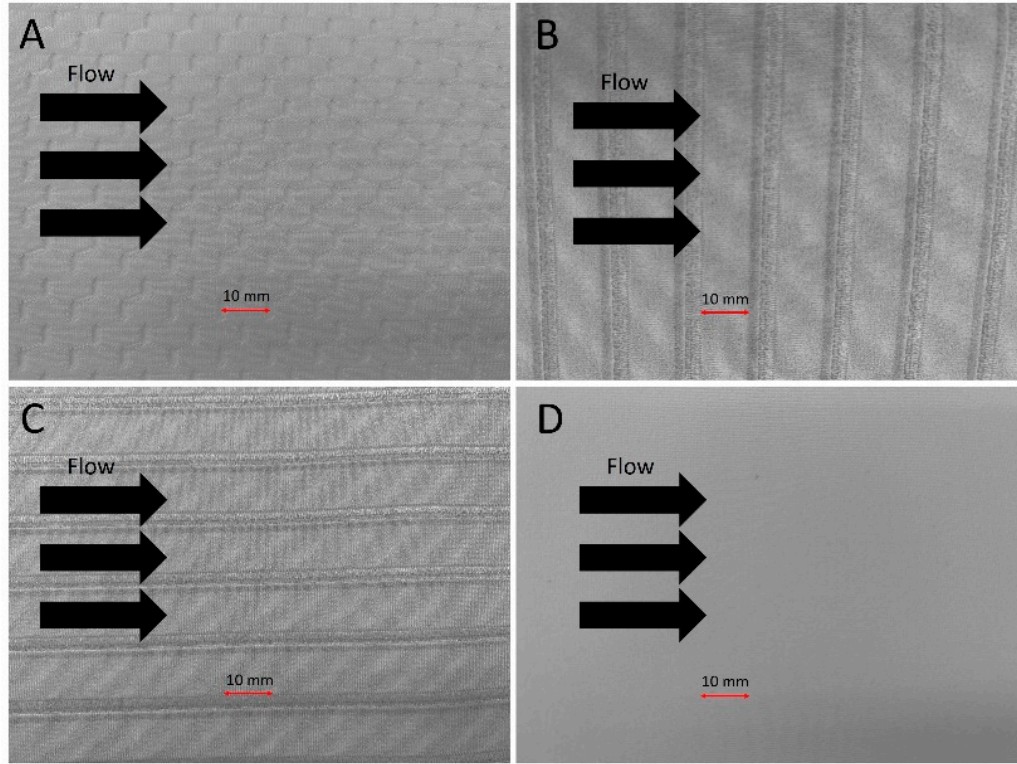

**Figure 2.** Cycling suit fabric types used in the wind tunnel experiment. ((**A**) has small uneven grooves. (**B**) has long and shallow vertical groove lines. (**C**) has horizontal groove lines. (**D**) is a smooth type of fabric.).

Moreover, based on the results of the fabric test conducted with the cylindrical model, the air resistance was compared using a full-scale mannequin. In this experiment, the arms of the actual mannequin were covered with the fabrics which were used in the cylinder model experiment (different types of fabrics with uneven surfaces with grooves) and the air resistance acting on the mannequin were measured with wind speeds between 57 and 61 km/h at 2 km intervals (Figure 3) and compared.

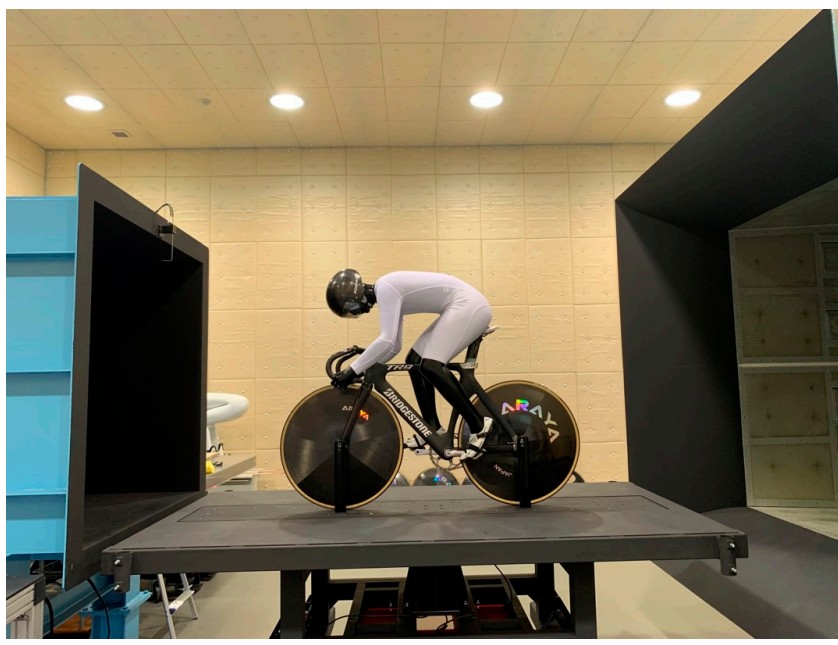

**Figure 3.** Full-scale mannequin used in the wind tunnel experiment.

### 2.2. CFD Analysis

The three-dimensional model shown in Figure 4 was created based on the full-scale mannequin used in the wind tunnel experiment. The analysis of this model was performed with CFD software (PowerFLOW5.5, Dassault Systèmes, Tokyo, Japan) based on the lattice Boltzmann method [18,19]. The lattice Boltzmann method is a simulation technique to model the thermal flow field of a fluid in which the fluid is approximated by a collection of virtual particles and the collisions and propagation of the particles and their moments are successively calculated using the velocity distribution function of the particles. More specifically, in order to compare how the airflow around the cyclist would change according to different wind speeds, the flow around the model was analyzed with two types of speed intervals (case 1: wind speed = 16 m/s, case 2: wind speed = 20 m/s) in this experiment. The objects of analysis were a racing bicycle and a cyclist wearing a helmet, gloves, skin suit and bike shoes. The posture of an actual cyclist was scanned with a 3D scanner and used for analysis. The crank angle was fixed at an angle parallel to the ground similar to the mannequin in the wind tunnel. In the calculations, the tire and wheel were treated as stationary with no rotation and the ground was treated as a stationary floor. The analysis was performed with the spatial grid resolution (VR9) at an offset of 10 mm from the actual shape. The analysis space (x, y, z) consists of areas of respectively 4.2 m, 0.8 m and 1.8 m and approximately 200 million cells with a standard size of 5 mm. Additionally, the standard cell size of the cyclist model surface is 1.25 mm. The physical conditions were obtained at a temperature of 20 °C and a representative density of 1.204 kg/m$^3$ under atmospheric pressure (Table 1).

**Table 1.** Analysis conditions.

| Item | Values | |
|---|---|---|
| Turbulence model | VLES (analyzed with Powerflow 5.5) | |
| Representative pressure [Pa] | 101,325 | |
| Representative temperature [°C] | 20.0 | |
| Representative density [kg/m$^3$] | 1.204 | |
| Representative kinematic viscosity [m$^2$/s] | $1.49 \times 10^{-5}$ | |
| Calculated physical time [sec] | 1.27 (case 1; 16 m/s) | 1.40 (case 2; 20 m/s) |
| Minimum voxel size [mm] | 1.25 | |
| Total voxels | 191.20 million | |
| Fine Equivalent Voxels (FEV) | 100.61 million | |
| Total surfels | 7.09 million | |
| FES (fine equivalent surfels) | 6.85 million | |

The behavior of the multi-particle motion system can be represented by the fundamental laws of mechanics that govern single-particle motions at the molecular scale. The Boltzmann equation can formulate a problem in terms of the distribution function $f(x, v, t)$, which indicates the molecular number density at position $x$, time $t$ and velocity $v$. This equation (in the absence of external forces) can be expressed as follows:

$$\frac{Df}{Dt} = \frac{\partial f(x, v, t)}{\partial t} + v \cdot \nabla f(x, v, t) = C(x, v, t) \tag{1}$$

In this equation, the total differential on the left-hand side represents the connective motion of the particles, while the right-hand side represents the complex intermolecular interactions (collisions). By integrating the distribution function, it is possible to obtain macroscopic variables such as the fluid density, velocity and pressure.

The main purpose of the collision operator is to orient the velocity distribution function to the equilibrium distribution. The Bhatnagar, Gross and Krook collision operator [20] was used here and can be defined as follows:

$$C(x,\ v,\ t) = -\frac{1}{\tau}\left[f(x,v,t) - f^{eq}(x,\ v,\ t)\right] \tag{2}$$

Assume that $\tau$ is the relaxation time of the fluid and $f^{eq}(x,\ v,\ t)$ is the equilibrium distribution function.

To solve these equations more efficiently, they were discretized on a three-dimensional cubic lattice using a D3Q19 model [20,21]. This model discretizes the given velocity space into 19 discrete velocities. The discrete Lattice Boltzmann equations that use a specific difference in time ($\Delta t = 1$) can be expressed as follows:

$$f_i(x + c_i\Delta t,\ t + \Delta t) - f_i(x,\ t) = C_i(x,\ t) \tag{3}$$

$$C(x,\ t) = -\frac{1}{\tau}\left[f_i(x,t) - f_i^{eq}(x,\ t)\right] \tag{4}$$

The volume boundary scheme was chosen as the fluid–structure interaction method. Here, the grain boundary conditions were applied on the surface itself (i.e., the facets that make up the geometry description). Each of these facets had a set of extruded parallelograms corresponding to the discrete velocity directions.

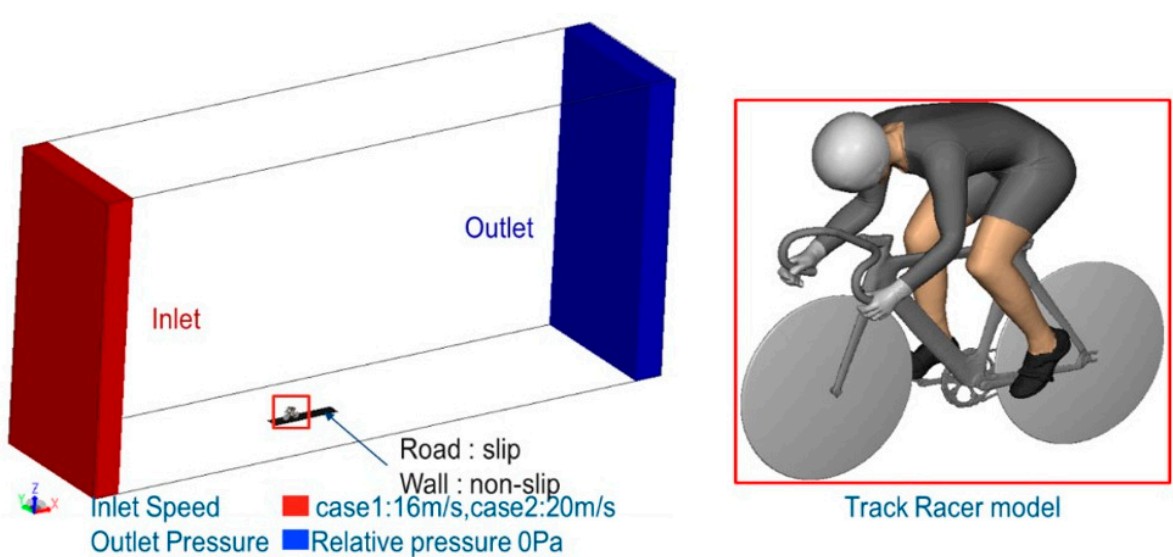

**Figure 4.** Calculation area boundary conditions and the 3-D model.

## 3. Results and Discussion

### 3.1. Visualization Results Using CFD

Figure 5 shows the pressure distribution on the cyclist represented by contour and flow lines. The simulation analysis was performed at speeds of 16 m/s and 20 m/s, which are common speeds in actual races. From the figure, it can be seen that the pressure is high at the regions that act as stagnation points for the flow of the model, such as the helmet as well as the region from the shoulders to the arms and thighs to the feet. It is likely that these body parts that are perpendicular to the traveling direction cause the flow to collide and stop moving, thereby increasing the pressure. Additionally, the difference in pressure between the left and right foot is attributed to the fact that while the right foot is stepping on the pedal from the front, the left foot is pulled backward. Moreover, higher speeds are believed to generate more pressure and, consequently, higher loads on the body.

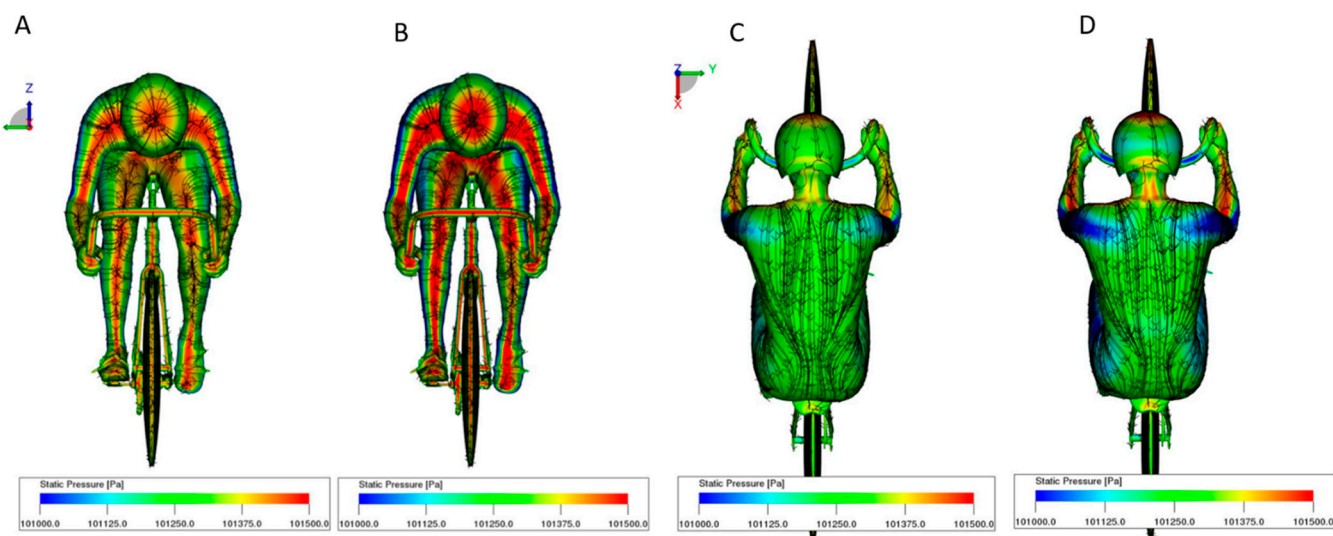

**Figure 5.** Time-averaged surface pressure distribution (contour and flow lines), (**A**,**C**) scheme 16 m/s (around 58 km/h) and (**B**,**D**) of 20 m/s (72 km/h).

In addition, when viewing from the top, it can be seen that the pressure is high not only at the top of the helmet and arms, which are perpendicular to the moving direction and are directly hit by the wind, but also at the rear of the helmet and base of the neck. This is attributable to the fact that the flow separates at the helmet and then curls around it, increasing the pressure at the neck, as shown in Figure 6.

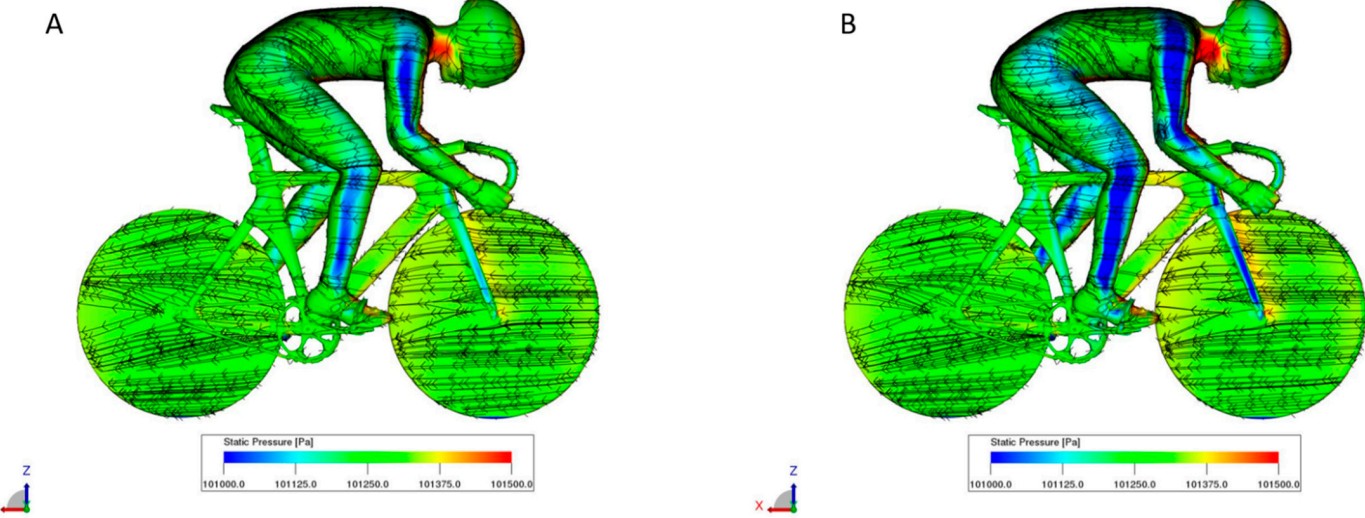

**Figure 6.** Side views of the time-averaged surface pressure distribution (contour and flow lines), (**A**) represents the model at 16 m/s and (**B**) at 20 m/s.

The side views in Figure 6 show that the pressure is lower on planes parallel to the traveling direction. In other words, it indicates that the pressure is lower at the side, on the cyclist's shoulder and in the area extending from the base of the foot to the ankle. In addition, this trend becomes even more evident with a faster speed of 20 m/s (Figure 6B). It can also be seen that the flow separates from the armpits and thighs to the buttocks. Two measures that could potentially reduce this air resistance were placing the stitches on the surface of the suit behind the separation point and using a fabric with a processed surface shape around the separation point.

Viewing the pressure distribution from the diagonally forward right side (Figure 7), it is noted that the pressure is also high in the abdominal area of the cyclist. However,

since this area is hidden by the upper body as shown in the frontal distribution diagram (Figure 5), this high pressure is attributed to the fact that the surrounding flow is stagnant and not because the area is directly hit by the flow. The pressure is also high at regions that act as stagnation points for the flow to the model, such as the helmet as well as the areas from the shoulders to arms and thighs to feet. On the other hand, the flow appears to follow the perimeter of the body without stalling on the side of the arms and feet, which are parallel to the traveling direction, thereby decreasing the pressure.

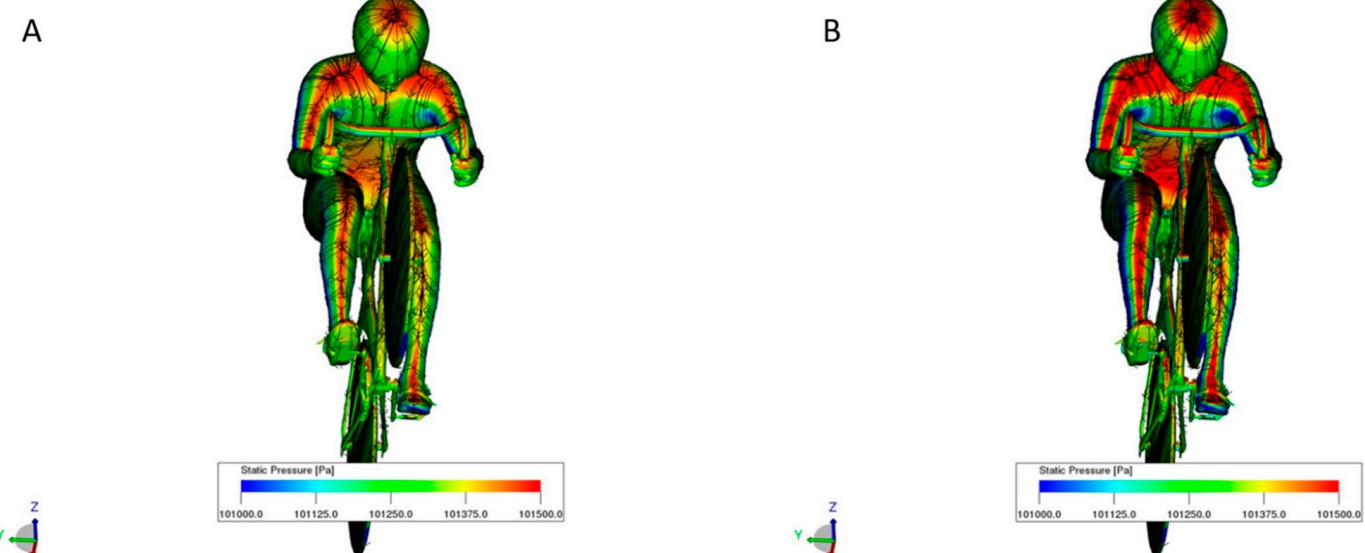

**Figure 7.** Time-averaged surface pressure distribution (contour and flow lines) seen from diagonally forward right, (**A**) represents the model at 16 m/s and (**B**) at 20 m/s.

### 3.2. Wind Tunnel Experiment Results

The CFD simulation results shown in Figures 5–7 demonstrate that the air resistance is high at the cyclist's head, arms and legs, which agrees with the results of previous studies [13,14]. We believed that an effective way to control the flow field at such areas and reduce air resistance was to conduct a basic experiment to measure air resistance using shapes that can simulate certain body parts, thereby creating an index for fabric selection for the research and development of suits. For the legs, however, it is necessary to examine the impact of the movement required to rotate the crank. Therefore, this study focused on how the fabric shape affects the air resistance by analyzing the basic aerodynamics using a cylinder model that simulates the arms of the cyclist (Figure 1).

Figure 8 shows the drag force results of the wind tunnel experiment using a cylinder model. Here, the wind speed is expressed in km/h instead of in terms of Reynolds numbers for easy comparison with previous results. As previously mentioned, four types of fabrics were used in this experiment: A, with an uneven surface; B, with vertical stripes (groove lines perpendicular to the wind flow); C, with horizontal stripes (groove lines parallel to the wind flow); and D, a smooth fabric (Figure 2). The results shown in Figure 8 N indicate that when the model is not wearing any suit (nude), the measured drag force increases as the speed increases. Additionally, in terms of the percentage of air resistance reduction relative to the nude model, the drag force measured at 60 km/h with fabric B (2.78 N) was reduced by approximately 25%. Moreover, a reduction in drag force was observed at all speed intervals (50–60 km/h) in case of fabric A and the air resistance was lower than that of the other fabrics. At the maximum speed interval (55–60 km/h), which is the maximum speed range achieved by the world's best cyclists in actual races, the result was particularly very good when the air resistance was seen to reduce by approximately 30%. Additionally, when fabrics C and D were used, the air resistance changed by approximately 1–2% with different wind speeds. These results suggest that the shape of the fabric surface

may change the airflow field around the cylinder according to the surrounding wind speed. Furthermore, they indicate that by selecting a material that best suits the cyclist's main speed, it is possible to generate propulsion in the most efficient manner.

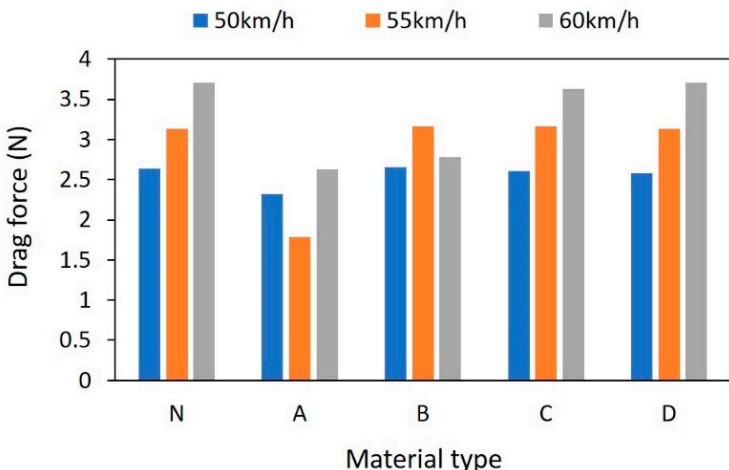

**Figure 8.** Comparison of drag forces of each fabric obtained from the cylinder test (N represents the drag force when the model was no fabric).

Figure 9 shows the measurement results of air resistance when the mannequin model was used. In the mannequin experiment, the intervals that showed larger differences in the cylinder test (55 or higher) were examined in further detail. To this end, in this mannequin test, we separated the wind speed into three different speeds ~57, 59 and 61 km/h and examined the respective aerodynamic forces. The drag coefficient ratio, Cd/Cd (Nude), represented in the vertical axis of Figure 9, is the value obtained by dividing the drag coefficient (*Cd*) obtained from the experimental results with the suit on by the drag coefficient Cd measured without the suit (Nude). Additionally, the AD suit type contains upper arms made of uneven fabric (fabric A) and the forearms made of fabric D. Moreover, the entire arms of BB and CC suit types are, respectively, made with striped fabrics B and C in the cylinder experiment, while the BD and CD types were, respectively, made with fabrics B and C in the upper arms and fabric D in the forearms, as with the suit AD (Figure 10).

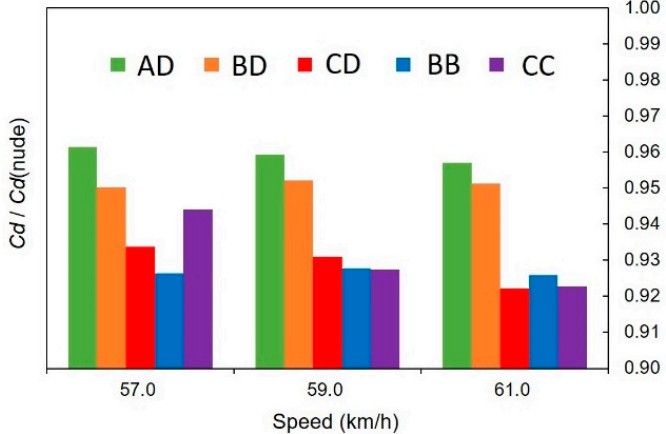

**Figure 9.** Comparison of drag coefficients of fabrics using the mannequin for bicycles. The number 1.00 on the vertical axis indicates the *Cd* value when the model is unclothed (nude) and the values in the graph are shown as a ratio with other fabric results.

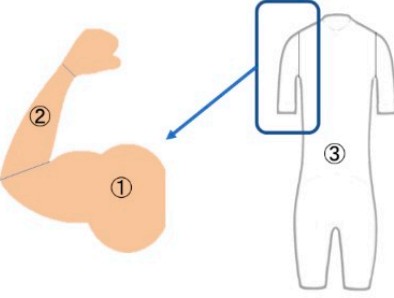

| Name | Material type | |
|------|------|------|
| | ① upper arm | ② forearm |
| AD | A | D |
| BD | B | D |
| CD | C | D |
| BB | B | B |
| CC | C | C |

**Figure 10.** Combinations of cycling suit fabrics used in this experiment. We separated the arms into upper arms and forearms and used five types of suits, made with combinations of four different types of fabric (A to D). The body part ③ of all the five suits was made with fabric D.

The experimental results shown in Figure 9 indicate that the *Cd* acting on the mannequin with the suit was lower than that without the suit and some fabrics used for the arms could reduce the air resistance acting on the mannequin even further. When viewing each suit individually, fabric C (with stripes), drastically reduced the *Cd* values at all speed intervals in the mannequin experiment. On the contrary, the drag reduction effect of A and B, which used materials that greatly reduced the air resistance value in the cylinder experiment, was smaller than that of fabric C in the mannequin experiment. For this reason, it is speculated that fabrics A and B may create a condition where it is more difficult to reduce air resistance on the entire mannequin with airflow from the upper arms to the back. In addition, in suit types BD and CD, in which the forearms are made of the same material as that in AD, air resistance is seen to decrease as the speed increases. This points to the possibility of adopting a fabric that suits the flow of the arms according to the traveling speed, thereby reducing the air resistance on the entire mannequin even further. The mannequin used in this experiment is different in many respects from a real cyclist; therefore, there may be physical changes in the fabric in actual scenarios owing to environmental changes such as body temperature and humidity, with unknown effects on the aerodynamics. Therefore, in future wind tunnel experiments, it may be necessary to improve the experimental setup. This may be realized, for example, by placing a hot light bulb on the mannequin to test the effect of different temperatures on the aerodynamics. Furthermore, in this experiment, it was difficult to directly compare the three models (cylinder model, mannequin model and CFD model) because the wind speeds were not identical. In subsequent experiments, experimental settings such as wind speed should be considered in more detail.

## 4. Conclusions

In this study, we conducted CFD simulations and a wind tunnel experiment to investigate the aerodynamics in track races as well as the movement of separation points around a cyclist, the airflow variation and how the fabric used in the suit affects air resistance. The results, which were in line with those of previous studies [8,13,14], indicated a high surface pressure on the cyclist's head, arms and legs as well as the occurrence of a flow that separates from the armpits, legs and lower back of the cyclist. Moreover, by changing the fabric pattern (shape) of the arms of the cycling skinsuit, it is possible to reduce the air resistance acting on the entire body of the cyclist by 4–8% compared to the nude model. Results of the experiment also confirmed that while some fabrics are highly speed-dependent and exhibit lower air resistance at higher traveling speeds, others led to an increase in air resistance. These findings suggest that a thorough understanding of the aerodynamics of skinsuit fabric considering the main speed interval used by cyclists, followed by an adequate application of this knowledge to the clothing structure, may lead to even better sports performance.

However, in this study, both the cylinder and mannequin models were made of resin and it is necessary to consider that in real-world scenarios the *Cd* (Nude) value may change when a cyclist wears a suit. In addition, when analyzing the actual cycling, it is important to consider the suit properties such as adhesion between skin and fabric, sweating and friction as factors, which will likely require further investigation and analysis with a broader set of experimental data in the future. In addition, the visualization results using CFD assumed a stationary model, not a bicycle in motion; therefore, the aerodynamic forces acting on each part were not compared. This aspect will be examined in future studies. Future wind tunnel experiments conducted on the cylinder model and mannequin should consider the effects of the different fabrics on airflow, such as the flow separation at the boundary layer around the model.

**Author Contributions:** Conceptualization, S.H. and T.A.; methodology, S.H.; software, S.H.; validation, S.H. and T.A.; formal analysis, S.H.; investigation, S.H.; resources, S.H. and T.A.; data curation, S.H. and T.A.; writing—original draft preparation, S.H.; writing—review and editing, S.H. and T.A.; supervision, S.H.; funding acquisition, S.H. and T.A. All authors have read and agreed to the published version of the manuscript.

**Funding:** This research was funded by the Ministry of Education, Culture, Sports, Science and Technology of the Japanese government, JSPS KAKENHI (Grant number: 20K11413, 20H04066).

**Institutional Review Board Statement:** Not applicable.

**Informed Consent Statement:** Not applicable.

**Data Availability Statement:** Data sharing not applicable.

**Acknowledgments:** This research has been partly supported by the Project: "Research and Development for low-resistance cycling skinsuit", funded by the DESCENTE LTD. (CFE29045) and Japan Keirin Autorace foundation (2018M-004).

**Conflicts of Interest:** The authors declare no conflict of interest.

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
