# Peer review of "Aerodynamics of Cycling Skinsuits Focused on the Surface Shape of the Arms"

_applsci, doi:10.3390/app11052200_

Round 1

Reviewer 1 Report

Please see attached review

minor corrections are to be brougth to the manuscript to clearify some questions

Author Response

Response to Reviewer 1 Comments

We have revised the paper following your guidance and comments as described below.

Abstract and introduction

Air resistance on bicycle and cyclist is estimated according to the data from reference 1 (Kyle &

Burke), a reference in the research field of the manuscript; can these data (90%, 70%) be completed and confirmed by later research papers ? A complete review paper, which should be cited in the manuscript, has been published recently by Fabio Malizia , Bert Blocken

Bicycle aerodynamics: History, state-of-the-art and future perspectives

Journal of Wind Engineering & Industrial Aerodynamics

https//doi.org/10.1016/j.jweia.2020.104134

The following content was added to the text:

[To be / added in page 1]

“Additionally, a review paper summarizing varied previous studies on bicycle aerodynamics was recently published.17

Section 2.

Wind tunnel

What is the free turbulence level (intensity) in the wind-tunnel flow

Line 71-73 sentence “the aerodynamics…” should be formulated differently since these four types of fabrics were never mentioned before in the text;

The following content was added to the text:

[To be / added in page 2]

“In this study, the four fabric types shown in Figure 2 were produced and compared in order to examine the aerodynamic effects of different surface shapes. Here, A has small uneven grooves, B has long and shallow vertical grooves, C has horizontal grooves, and D is a smooth type of fabric.”  

What does it mean in terms of surface roughness for an uneven surface (type A), why choose stripe distance of 10 mm (figure 2) ?

The following content was added to the text:

[To be / added in page 2]

“The groove lines were spaced 10 mm apart. As basic research on different types of fabrics, this study entailed an investigation of the aerodynamic properties of these four fabric types in the wind tunnel.”  

Subsection 2.2

Using the transient Lattice Boltzmann-based physics powerFLOW from Dassault seems a good

choice, some details should be given (with reference, not only reference 18) on the VLES turbulence model and the computer system used for the calculations, also in terms of time costs for the simulations.

The following content was added to the reference:

Section 3.

Visualizations

It has to be pointed out the pressure distribution is calculated on a static mannequin, which does

Not correspond to dynamic race conditions.

The following content was added to the text:

[To be / added in page 4]

“In the calculations, the tire and wheel were treated as stationary with no rotation, and the ground was treated as a stationary floor.”  

149-153 : the wake of the helmet strongly depends on its geometry, sentence “Presumably…” should be formulated differently.

We have modified text as per your comment.

[To be / added in page 6]

“This is attributable to the fact that the flow separates at the helmet and then curls around it, increasing the pressure at the neck, as shown in Figure 6.”  

Can the contributions of pressure to drag force been quantified on the different surfaces (helmet,

shoulders, arms, body, legs) ?

The following content was added to the text:

[To be / added in page 9]

“The mannequin used in this experiment is different in many respects from a real cyclist; therefore, there may be physical changes in the fabric in actual scenarios owing to environmental changes such as body temperature and humidity, with unknown effects on the aerodynamics. Therefore, in future wind tunnel experiments, it may be necessary to improve the experimental setup. This may be realized, for example, by placing a hot light bulb on the mannequin to test the effect of different temperatures on the aerodynamics.”  

[To be / added in page 10]

“In addition, the visualization results using CFD assumed a stationary model, not a bicycle in motion; therefore, the aerodynamic forces acting on each part were not compared.”  

Why choose 20 m/s (72 km/h) for computations if it is compared to the speeds used in the wind

tunnel lower than 61 km/h ?

The following content was added to the text:

[To be / added in page 6]

“The simulation analysis was performed at speeds of 16 m/s and 20 m/s, which are common speeds in actual races.”  

Figure 8

For the studies concerning the cylinder, it would be appropriate to make the analysis in terms of

Reynolds numbers and take a look at the literature (and a reference or two) and reports on the drag coefficient and wake of a cylinder at different regimes (Reynolds numbers) for smooth or rough surfaces. And verify if you can be in the regime of drag crisis (Eiffel paradox) for case A or B.

The following content was added to the text:

[To be / added in page 7]

“Here, the wind speed is expressed in km/h instead of in terms of Reynolds numbers for easy comparison with previous results.”  

Further information should be eventually given on the wake behind the cylinder, boundary layer separation line ?

The following content was added to the text:

[To be / added in page 10]

“Future wind tunnel experiments conducted on the cylinder model and mannequin should consider the effects of the different fabrics on airflow, such as the flow separation at the boundary layer around the model.”  

For case B can you indicate if a groove line is perpendicular to the flow (in the yz plane) ?

We have modified text and Fig. 2 as per your comment.

[To be / added in page 8]

“As previously mentioned, four types of fabrics were used in this experiment: A, with an uneven surface; B, with vertical stripes (groove lines perpendicular to the wind flow); C, with horizontal stripes (groove lines parallel to the wind flow); and D, a smooth fabric (Figure 2).”  

Line 198 : 55-60 km/h is in the range of the word records per hour, this can be mentioned to.

We have modified text as per your comment.

[To be / added in page 8]

“At the maximum speed interval (55–60 km/h), which is the maximum speed range achieved by the world's best cyclists in actual races,”  

Figure 10

It is difficult to compare and extrapolate the results with figure 8 since the speeds (especially 50 &55km/h) are not exactly the same and the experimental tests are very sensitive for some fabrics in these speed ranges; nevertheless it is interesting to see that cases CD and BB reduce drag by

several % and case CC is interesting for high speeds (59-61 km/h).

Mannequin modeling

16 m/s & 20 m/s (57.6 km/h & 72 km/h)

Mannequin drag experiments

57, 59, 61 km/h (15.8 m/s, 16.4 m/s &16.9 m/s)

Cylinder drag experiment : 13.88 m/s (50 km/h) to 16.66 m/s (60km/h)

Re = 93 000 to Re = 112 000

The following content was added to the text:

[To be / added in page 10]

“Furthermore, in this experiment, it was difficult to directly compare the three models (cylinder model, mannequin model, and CFD model) because the wind speeds were not identical. In subsequent experiments, experimental settings such as wind speed should be considered in more detail.”  

Reviewer 2 Report

Review of the manuscript entitled: Aerodynamics of cycling skinsuits focused on the surface shape of the arms.

The reviewed manuscript is very interesting. FSI simulation is an interesting and relatively difficult approach for modeling. On the other hand, the same situation concerns experimental validation of the proposed model. Generally, the manuscript is written well (as a synthesis of a project report), however contains some elements that need to extend. In the present form, the manuscript is understood by a research team member or subject specialist.
For the potential reader, the paper should explain some methodology and measurement elements.
Below the main elements that should need for extended or explain:
1. Lattice-Boltzmann discretization should be described in more detail.
2. The experimental (extensometers?) measurement should be described in more detail.
3. The FSI approach should be described in more detail.
4. The choice of the weave and material of the cycling skinsuit should be described in greater detail and justified.
5. Initial conditions and simplifications of the case (simulation model) should be detailed.
6. Prognosed influence of adhesion between skin and fabric and sweating will important in view of obtained results. So, some extended discussion about it is a necessary, not only laconic sentence about the future research direction in the conclusions.
In conclusion: I recommended a major revision of the manuscript as a worth study from the point of view of the current and currently developed studies, especially as the example of the Lattice-Boltzmann discretization and FSI approach.

Author Response

Response to Reviewer 2 Comments

We have revised the paper following your guidance and comments as described below.

The reviewed manuscript is very interesting. FSI simulation is an interesting and relatively difficult approach for modeling. On the other hand, the same situation concerns experimental validation of the proposed model. Generally, the manuscript is written well (as a synthesis of a project report), however contains some elements that need to extend. In the present form, the manuscript is understood by a research team member or subject specialist.

For the potential reader, the paper should explain some methodology and measurement elements.
Below the main elements that should need for extended or explain:

  1. Lattice-Boltzmann discretization should be described in more detail.
  2. The experimental (extensometers?) measurement should be described in more detail.
  3. The FSI approach should be described in more detail.
  4. The choice of the weave and material of the cycling skinsuit should be described in greater detail and justified.
    5. Initial conditions and simplifications of the case (simulation model) should be detailed.
    6. Prognosed influence of adhesion between skin and fabric and sweating will important in view of obtained results. So, some extended discussion about it is a necessary, not only laconic sentence about the future research direction in the conclusions.

The following content was added to the text:

[To be / added in page 4]

“The lattice Boltzmann method is a simulation technique to model the thermal flow field of a fluid in which the fluid is approximated by a collection of virtual particles, and the collisions and propagation of the particles and their moments are successively calculated using the velocity distribution function of the particles. More specifically, in order to compare how the airflow around the cyclist would change according to different wind speeds, the flow around the model was analyzed with two types of speed intervals (case 1: wind speed= 16 m/s, case 2: wind speed= 20 m/s) in this experiment. The objects of analysis were a racing bicycle and a cyclist wearing a helmet, gloves, skin suit, and bike shoes. The posture of an actual cyclist was scanned with a 3D scanner and used for analysis. The crank angle was fixed at an angle parallel to the ground similar to the mannequin in the wind tunnel. In the calculations, the tire and wheel were treated as stationary with no rotation, and the ground was treated as a stationary floor. The analysis was performed with the spatial grid resolution (VR9) at an offset of 10 mm from the actual shape. The analysis space (x, y, z) consists of areas of respectively 4.2 m, 0.8 m, and 1.8 m and approximately 200 million cells with a standard size of 5 mm. Additionally, the standard cell size of the cyclist model surface is 1.25 mm. The physical conditions were obtained at a temperature of 20 °C and a representative density of 1.204 kg/m3 under atmospheric pressure (Table 1).”

[To be / added in page 9]

“The mannequin used in this experiment is different in many respects from a real cyclist; therefore, there may be physical changes in the fabric in actual scenarios owing to environmental changes such as body temperature and humidity, with unknown effects on the aerodynamics. Therefore, in future wind tunnel experiments, it may be necessary to improve the experimental setup. This may be realized, for example, by placing a hot light bulb on the mannequin to test the effect of different temperatures on the aerodynamics. Furthermore, in this experiment, it was difficult to directly compare the three models (cylinder model, mannequin model, and CFD model) because the wind speeds were not identical. In subsequent experiments, experimental settings such as wind speed should be considered in more detail.”

Round 2

Reviewer 2 Report

Re-review of the manuscript entitled: Aerodynamics of cycling skinsuits focused on the surface shape of the arms

Re-review of the manuscript entitled: Aerodynamics of cycling skinsuits focused on the surface shape of the arms
Like I found the manuscript is corrected. The authors prepared adequate answers,  provide corrections, and improve the manuscript. The reviewed paper has very interesting applied aspects. FSI simulation is an interesting and relatively difficult approach for modeling. On the other hand, the same situation concerns experimental validation of the proposed model discretized by the Lattice-Boltzmann approach. The subject is an actual, so the paper should be interesting for the potential readers and should be citable also in the future. In summary: in view of responses and prepared corrections, I recommend accepting the paper in the present form.